# The Effectiveness of Management Ability on Firm Value and Tax Avoidance

**Maryam Seifzadeh** 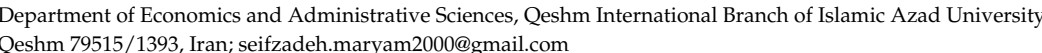

Department of Economics and Administrative Sciences, Qeshm International Branch of Islamic Azad University, Qeshm 79515/1393, Iran; seifzadeh.maryam2000@gmail.com

**Abstract:** The current study investigates the relationship between tax avoidance, management ability, and firm value. Three hypotheses are proposed to meet the paper's objective. For conducting such a practical study based on a post-event descriptive correlational approach, data are gathered from the website of the Tehran Stock Exchange during 2014–2020. A total of 183 companies were selected through the systematic elimination method and analyzed using the R statistical software. The results indicated a negative relationship between managerial ability and tax avoidance. Moreover, we find a significant negative relationship between tax avoidance and firm value. Finally, the findings argue that in companies with high-ability managers, the intensity of the negative relationship between tax avoidance and firm value is mitigated.

**Keywords:** tax avoidance; managerial ability; firm value

## 1. Introduction

Managers are responsible for making significant strategic decisions and planning operations for companies. In particular, under rapid changes and intense competition in a business setting, managers' different formulated management strategies could shape a company's future and contribute significantly to the firm's value. Tax avoidance is a managerial decision (Al-Maliki et al. 2022). Managers can allocate specific resources to the company; otherwise, the authorities may tax them. In contrast to other management activities, tax avoidance benefits derive from a reservoir with a decrease in tax costs. However, tax avoidance activities would carry some non-tax costs, including direct costs for necessary tax strategies, financial reporting, agency, political, and stigma-related costs (Park et al. 2016; Salehi et al. 2022).

Most previous studies indicated that manager characteristics could affect a firm's economic consequences, affecting the economy, accounting, and management significantly, but influencing the business methods (Bertrand and Schoar 2003; Gabaix and Landier 2008). However, the experimental analysis of management activities related to managerial ability is complex because realizing a company's and manager's characteristics is cumbersome (Chang et al. 2010). Previously, the obtained unnatural returns from asset or stock returns, the degree of pressure, or the effects of managers' characteristics were used as the indexes of managerial ability (Salehi et al. 2021). Such managerial ability metrics involve major firm characteristics out of control (Seifzadeh et al. 2022). For example, large corporations cope with more pressures than smaller ones, and the asset or stock return could be affected by various factors (Khelil and Khlif 2022).

Although the effects of a manager's characteristics could be peculiar to that manager, they could be functional only in limited cases where the manager works on more than one project (Ko et al. 2013). Moreover, numerous studies analyzed the relationship between tax avoidance and manager characteristics. Dyreng et al. (2010) revealed that a manager's characteristics could affect effective tax rate changes. Still, such effects cannot determine the relationship between tax avoidance and a manager's specific characteristics, including education, age, gender, and CEO tenure.

This study examines how tax avoidance is derived from managerial ability. Demerjian et al. (2012) state that managerial ability increases a firm value by using limited resources effectively throughout the business operation. In this study, managerial ability measurement is determined by the firm's efficiency after controlling the company's factors that affect the managers. In addition, such a measurement can be changed over time, while a manager's characteristics are fixed, like experiences. Finally, the present study attempts to show experimentally that managerial ability is one of the factors determining the strategies of a firm's tax avoidance after controlling the characteristics of a firm. Moreover, since different studies have mixed results concerning the relationship between tax avoidance and firm value, important research questions are raised concerning how tax avoidance is affected by managers' ability, who take firm value into account as a top priority. Accordingly, we describe the theoretical literature and the required mechanisms to implement the objectives and test the research hypotheses in the following.

## 2. Theoretical Framework

A comprehensive stream of investigations explores the effect of managers on firms' decision-making processes. For example, some examine the managers' influence on a firm financial policy and investment decisions, such as dividends, capital expenditures, and mergers and acquisitions (Bertrand and Schoar 2003). Moreover, managerial ability is defined as increasing the firm value using limited resources throughout the business operation (Demerjian et al. 2012). Managers with a higher ability level are expected to have more awareness and understanding of the industry. Demerjian et al. (2013) evaluate the impact of individual managers on earnings quality with the measure of managerial ability. They reveal that managerial ability is positively (negatively) related to accruals quality (restatements). High-ability managers are expected to make more efficient, sophisticated judgments and assess the reflection of firms' transactions. Similarly, the managerial style literature mainly involves managers who replace multiple firms; using managers' fixed effects, they measure the effect of managers' characteristics on firms' decision processes. These studies examine the managerial style of voluntary information disclosure and earnings quality (Bamber et al. 2010; Dejong and Ling 2013). Dyreng et al. (2010) show a manager-specific effect on corporate tax avoidance using a manager fixed effects research design. In this study, employing Demerjian et al. (2012) measurement, which estimates the managers' ability to manage limited resources efficiently, we attempt to capture the managers' fixed characteristics explained by tax avoidance.

The reduction of explicit taxes is the widely accepted definition of tax avoidance. More precisely, tax avoidance consists of tax savings from the companies' activities (Salehi and Salami 2020). Tax programming through realization, capitalization, and unique benefits through political communications and lobbying (Hanlon and Heitzman 2010). However, the corporate tax avoidance study focuses on firm characteristics as determinants (Gupta and Newberry 1997; Mills 1998; Rego 2003; Wilson 2009); in contrast, the research considers the potential role of individual corporate decision-makers on firms' tax avoidance strategies, are limited.

Considering the above discussions, assuming a similar situation of other factors, managers with a higher ability to allocate resources efficiently may be expected to be involved in higher tax avoidance (Salehi et al. 2020a). There are three underlying reasoning for such a prediction. First of all, managers with more ability may make more efficient financial decisions in line with tax strategies and can identify and design tax planning processes effectively due to their deeper perception of the company's operating atmosphere under their management (Akbari et al. 2019). For instance, time management, classifying the financial transactions such as research and development (R&D) activities, which have prominent contributions to gain tax benefits by R&D tax credits, and structuring the managers and acquisitions (M&A) in ways that create more significant tax benefits. Secondly, more able managers are expected to concentrate on cost reduction. Most managers show a propensity for cost reduction, but only those with a higher ability to efficiently allocate firm resources

may significantly reduce cost (Akbari et al. 2018). It is noticeable that a reduction in operating costs, including marketing, production, labor, and R&D, may impact firm operations adversely. For example, reducing sales and a firm's innovation may result from marketing expenses, R&D spending, and employee salary cuts. However, it is assumed that cutting tax costs is less likely to affect firm operations adversely. All managers are looking to allocate resources efficiently to reduce costs through tax avoidance. Thirdly, since cash flows can be applied in firm operations to generate a positive return on investment, high-ability managers, who allocate their firm's resources effectively, are more likely to make financial decisions in line with decreasing income tax cash outflows, according that every paid cash tax is a unit of cash flow, which cannot add value within firm's operation. In other words, the reallocation of firm resources, from tax payments, into operating activities, is likely to attract managers with a higher ability to allocate firm resources effectively. Generating cash flows through tax avoidance is applicable in firms (Armstrong et al. 2015; Salehi and Salimi 2017). Guan et al. (2018) find that managerial ability influences the formation of tax-efficient dividend policies. Koester et al. (2016) find that, through engagement in greater state tax planning activities, managers attempt to shift more income to foreign tax havens, make more R&D credit claims, and make greater investments in assets that generate accelerated depreciation deductions. Additionally, they show that manager characteristic related to firms' tax policy decisions adds to our understanding of the factors that explain the substantial variation in corporate income tax payments across firms. Li et al. (2018) argue that the inevitable disclosure doctrine recognition increases the cost of job loss for managers whose current jobs are in jeopardy, thereby increasing their incentives to avoid taxes, and improve performance. Banker et al. (2018) find that managerial ability improves the future program expense ratio. Huang and Zhang (2019) find that financial expert CEOs are associated with a more aggressive tax avoidance policy. Further, their results indicate that the impact of financial expert CEOs results from a careful analysis of cost and benefit.

However, managerial ability might play a significant role in reducing a firm's cost. Still, several obstacles do not let high-ability managers avoid more income tax than low-ability managers in their industry peers. Hanlon and Heitzman (2010) argue that one potential factor affecting tax avoidance is the firm characteristics due to managers' prior strategic decisions. Since making new strategic decisions, changing a firm's operations, altering R&D strategies, etc., may charge additional costs to firms, which reduces managers' options in tax avoidance activities. For instance, firms may operate in foreign countries to reduce production costs or find a broader range of suppliers and customers. In some cases, operating overseas allows firms to pay lower tax rates on their income obtained in foreign countries, which can be considered a tax advantage. Establishing a foreign branch or subsidiary without extensive foreign manufacturing, suppliers, or customers may not benefit companies to avoid taxes. In another stream of study, previous literature demonstrates that other incentives might negatively influence tax avoidance policies (Rego and Wilson 2012). Moreover, there is the possibility that high-ability managers are likely to concentrate on business operations, which in turn profoundly makes tax avoidance strategies their second priority concern. Park et al. (2016) showed that managerial ability and tax avoidance are negatively related. Managers with more remarkable abilities are less likely to employ tax avoidance designs. Akbari et al. (2018) expect a negative association between tax avoidance and managerial ability. They show that there is no statistically significant association between these two variables. In addition, different levels of management ownership are associated with varying levels of tax avoidance (Cabello et al. 2019).

In line with the advocates of the opposite view, we collectively expect that high-ability managers may not always employ tax avoidance strategies in their management styles. Therefore, the first hypothesis is developed as follows:

**Hypothesis 1 (H1).** *Managerial ability has a negative impact on employing tax avoidance strategies.*

Generally, two views explain the relationship between tax avoidance and firm value named traditional and agency theory views. On the one hand, it is also known as the traditional view, proposing that owners' wealth is likely to be increased by designing tax avoidance strategies to decrease the company's cash outflow. In other words, tax avoidance may save financial resources that must be taken by the tax authority otherwise. Should there be no additional costs, which are named above, to avoid paying tax and no legal risk, the cash flow and income will increase by employing tax avoidance strategies. Some managers are motivated to engage in tax avoidance strategies actively. Such an explanation supports the idea that managers do not take action against the shareholders' interest in the process of tax avoidance. On the other hand, the agency theory view considers the positive points of tax avoidance, such as the decreased outflow of cash. It feels the direct and indirect costs of tax avoidance. The proposed direct costs comprise conducting firm resources for tax designation, such as employing expert staff or establishing a tax department, providing an information system for taxation, replacing the location of the company or its operations, increasing tax consultants, and lobbying for tax advantages (Mills et al. 1998; Lynch 2014; Brown et al. 2015). They named several indirect costs related to such strategies, such as political costs, reputational risks, financial reporting qualities, higher taxation, legal risks, and financial statement auditor scrutiny (Zimmerman 1983; Graham et al. 2014; Mills 1998; Frank et al. 2009; Hoopes et al. 2012).

The agency theory also argues that by disclosing less tax information, it is hard for investors and creditors to perceive the nature of tax avoidance behavior and evaluate whether it is willing to increase the firm's value or not (Desai et al. 2007) which in turn raise the expectation of outside investors for relatively higher returns (Kang and Ko 2014). In contrast, tax avoidance strategies increase the likelihood of limited disclosure to avoid the tax authorities' conjectures (Desai and Dharmapala 2008). Managers may employ tax avoidance strategies since it provides them with short-term interests. In other words, the discrimination between ownership and management opens an environment in which managers might act in line with their personal interests considering tax avoidance. In such a case, shareholders are expected to conduct efficient controls, such as compensation, to reduce agency costs (Jensen and Meckling 1976). Suppose the ambiguous financial reports have led to information asymmetry, and compensation contracts do not mitigate the knock-on effect of managerial self-interest. In that case, managers will act against the shareholders' interests. It is also known as an unethical firm by investors, creditors, and society. Therefore, the outcome and consequences of tax avoidance outweigh its benefits, resulting in decreased firm value (Hanlon and Slemrod 2009; Kang and Ko 2014; Son et al. 2012).

Prior literature shows a negative relationship between tax avoidance and firm value (Park et al. 2016). Sikes and Verrecchia (2016) showed that the cost of capital increases as firms engage in avoidance. Jacob and Schütt (2019) show that two dimensions of tax avoidance, uncertainty and the level of expected future tax rates, are jointly related to firm value and need to be expressed as a ratio. Hutchens et al. (2019) claim that tax avoidance strategies result in greater after-tax cash flows. Considering the above discussion, we expect that tax avoidance may burden further risk on firms at the charge of firm value. Therefore, the second hypothesis is conducted as follows.

**H2.** *Employing greater tax avoidance strategies decreases the firm value.*

According to conventional wisdom, managers are responsible for planning firms' strategies, which are expected to increase the firms' value through utilizing limited resources in business operations (Demerjian et al. 2012). Tax avoidance is a part of managerial decisions. Managers with high ability are more informed of their business settings and related industry to maximize efficiency through optimal resources (Park et al. 2016). Slemrod (2004) believes that the efficient level of tax avoidance might increase the firm value, which is different based on the firm's characteristics.

Park et al. (2016) indicate that higher managerial ability decreases the negative relationship between tax avoidance and firm value. Khurana et al. (2018) find that as tax avoidance increases, firms with high (low) managerial ability exhibit increased (reduced) investment efficiency. Salehi et al. (2020b) suggest that competition discourages managers from investing in risky investments. Moreover, managerial ability does not affect the association between product market competition and investment decisions. Ratu and Siregar (2019) show significant positive evidence of the effect of environmental uncertainty on tax avoidance. Gul et al. (2018) showed a significant association between firm acquisition decisions and corporate tax avoidance. Lee and Yoon (2020) found a significantly positive relationship exists between managerial ability and the tax cost variable. Simamora (2021) indicates that risk-taking behavior positively affects firms' performance for higher managerial ability. Baik et al. (2018) find a positive relationship between managerial ability and a firm's information environment by improving disclosure quality. Therefore, we developed the following hypothesis to answer whether managers have any role in rectifying the negative consequences of tax avoidance on the firm's value.

**H3.** *Managers can mitigate the negative association between tax avoidance and firm value.*

## 3. Research Methodology

The present study is descriptive correlational, and the statistical model used in this project is the multi-variation regression model. The required dependent, independent, and control variables were gathered from the financial statements of listed companies on the Tehran Stock Exchange via the official website of the Securities and Exchange and Rah-Avard Novin Software.

### 3.1. Data Calculation and Collection

The following regression model is used to test the first hypothesis based on the study of Park et al. (2016):

$$\text{TAXAVOID}_{it} = \beta_0 + \beta_1 \text{MA}_{it} + \beta_2 \text{ROA}_{it} + \beta_3 \text{LEV}_{it} + \beta_4 \text{NOL}_{it} + \beta_5 \text{PPE}_{it} + \beta_6 \text{INTAN}_{it} + \beta_7 \text{SIZE}_{it} + \beta_8 \text{MTB}_{it} + \beta_9 \text{AGE}_{it} + \beta_{10} \text{R\&D}_{it} + \beta_{11} \text{INDUSTRY} + \varepsilon_{it} \tag{1}$$

For the significance of the first hypothesis, β1 should be significant in the so-called model. The following regression model is used to test hypotheses 2 and 3 based on the study of Park et al. (2016):

$$\text{FIRMVALUE}_{it} = \beta_0 + \beta_1 \text{TAXAVOID}_{it} + \beta_2 \text{MA}_{it} + \beta_3 \text{TAXAVOID*MAhigh}_{it} + \beta_2 \text{ROA}_{it} + \beta_3 \text{LEV}_{it} + \beta_4 \text{NOL}_{it} + \beta_5 \text{PPE}_{it} + \beta_6 \text{INTAN}_{it} + \beta_7 \text{SIZE}_{it} + \beta_7 \text{GROWTH}_{it} + \beta_8 \text{AGE}_{it} + \beta_9 \text{R\&D}_{it} + \beta_{10} \text{OCF}_{it} + \beta_{11} \text{INDUSTRY} + \varepsilon_{it} \tag{2}$$

According to the study by Park et al. (2016), for the significance of the second and third hypotheses in this model, the β3 and β1 coefficients should be significant, respectively.

### 3.2. Dependent Variables

TAXAVOID$_{it}$: tax avoidance is calculated according to the residual of the following regression model. This variable is dependent on the first model of the study.

$$\text{BTD/ASSET}_{i,t-1} = \beta_0 + \beta_1 \text{TA/ASSET}_{i,t-1} + \varepsilon_{it} \tag{3}$$

where,

BTD$_{it}$: is the book-tax difference, income taxable difference is achieved in the company i in the year t with tax revenue divided by the total assets of the company i at the beginning of period t. income revenue is also achieved by dividing definitive diagnostic tax by the finance department into the ownership rate (22.5%).

ASSET: total book value of assets of the company i in the year $t-1$.

TA: discretionary accruals, for the calculation of which we have:

Operational cash-operational profit = discretionary accruals.

FIRMVALUEit is the firm value, a dependent variable in the second model and measured through Tobin's Q variable. To measure the firm value, we have:

$$\text{Tobin's Q} = \frac{\text{MVOCE} + \text{BVOLTD} - (\text{BVOSHTA} - \text{BVOSHTL})}{\text{BVOTA}} \quad (4)$$

where

MVOCE: is the market value of the ordinary stock at the year-end;
BVOLTD: is the book value of long-term debts at the year-end;
BVOSHTA: is the book value of current assets at the year-end;
BVOSHTL: is the book value of current debts at the year-end; and
BVOTA: is the book value of total assets at the year-end.

### 3.3. Independent Variables

MAt: Demerjian et al. (2012) proposed two criteria for measuring management ability. They declared that the firm resources available to the management include the cost of sold goods, office and sales costs, research and development costs, net fixed assets, and intangible assets. The output of applying these resources is the sales revenue. The following equation measures the total efficiency of the firm resources:

$$\max_v \theta = \frac{\text{Sales}}{v_1\text{CoGS} + v_1\text{R\&D} + v_1\text{SG\&A} + v_1\text{PPE} + v_1\text{OtherIntan}} \quad (5)$$

where

CoGS: is the final price of goods sold;
R&D is the cost of research and development;
SG&A: is sales and office cost;
PPE: is net fixed assets; and
OtherIntan: is intangible assets

First, the firm efficiency (MAX $\varepsilon$) is calculated through the data envelopment method. According to the guideline of Demerjian et al. (2012), in the first step of efficiency calculation, the firm's efficiency score is calculated as the efficiency criterion using the firm sales (output), the final price of goods sold, intangible assets, research and development costs, sales and office costs, and net fixed assets (inputs) and using the DEAP14 Software for each firm. Number 1 is used for highly efficient firms, and smaller figures (to 0) are related to low-efficiency companies. Demerjian et al. (2012) argued that the total efficiency of a firm, which is calculated by the equation mentioned above, shows the efficiency of resources available to the management as well as the personal capabilities of the management because a highly competent manager, regardless of the firm size, is more effective for predicting firm's mechanisms and procedures and for negotiating with major customers and suppliers. In addition to management ability, the resultant total efficiency depends on factors such as firm size, market share, free cash flows, number of branches or marginal units (the complication of the operation), and currency index (as a factor derived from foreign relations). Such a regression relationship is shown in the model (7–3) as follows:

$$\text{Firm Efficiency it} = \alpha 0 + \beta 1 \text{SIZE it} + \beta 2\ \text{MS it} + \beta 3\ \text{FCF it} + \beta 4 \text{Ageit} + \beta 5 \text{BUSEGit} + \varepsilon\ \text{it} \quad (6)$$

where

SIZE: is the natural logarithm of total sales of the company i in the year t;
MS: is the market share of the sales ratio of the company i in the year t to industry sales in the year t;
FCF: free cash flow, which is the operational profit before the depreciation minus tax cost, the interest cost, and share benefit paid to ordinary shareholders, which is calculated by dividing the book value of total assets of the company I in the year t;
Age: firm age;

BUSEG: is marginal firms' sales ratio (dependent) to total sales of the company i in the year t. If the company is not the dependent unit, it is assumed to be 1.

ε it: is the residual of the above model, which according to Demerjian et al. (2012), is the same management ability of company i in the year t.

*3.4. Control Variables*

The control variables of this study are as follows:

SIZEit: is the natural logarithm of total sales of the company i in the year t (to control the effect of the firm size);

ROAit: is the return of assets of the company i in the year t, which is the net profit ratio before interest and tax to the total market value of assets (to control the performance of the asset);

MTBit: is the market value ratio of dividends to its book value in the company i in the year t (to control the firms' growth opportunities);

LEVit: is the financial leverage, that is, debts book value ratio to the market value of total assets in the company i in the year t (to control the financial leverage or debt repayment ability);

AGEit: is the natural logarithm of the age of the company i.

NOLit: is indicative of losing companies, which is a dummy variable and is 1 if the company is losing; otherwise 0;

PPEit: is the fixed net assets ratio to book value of total assets of the company i in the year t;

INTANit: is the intangible assets ratio to book value of total assets of the company i in the year t;

RDNit: is the costs of research and development to book value of total assets of the company i in the year t;

GROWTHit: sales growth of the company i in the year t in proportion to the previous year;

OCFit: operational cash flows to book value of total assets in the company i in the year t;

YEAR: in order to control the years under study; and

INDUSTRY: to control the industry under study by the homogeneity of selected industries, the present industry groups in the Securities and Exchange were classified into seven subgroups from the Tehran Stock Exchange, as depicted in the previous table.

The statistical population of this study includes all listed companies on the Tehran Stock Exchange. The manufacturing companies understudy in the Tehran Stock Exchange with the following qualifications were selected as the statistical sample of the study:

1.　Due to the urgency of data covering the study's course, the name of companies under study should be inserted in the list of manufacturing companies on the Tehran Stock Exchange before 20 March 2014.
2.　They should present the data to the Securities and Exchange for at least seven years, from the beginning of 2014 to the end of 2020.
3.　They should be active during the study, transacted shares, and have no trading interruption.
4.　They should be affiliated with investment and intermediary financial companies.
5.　Given the duration of the study and by applying the conditions mentioned above, a total of 183 companies were selected as the study's sample. For the homogeneity of the selected industries for performing the calculations of discretionary accruals, the Tehran Stock Exchange industry groups were classified into seven subgroups.

## 4. The Research Findings

*4.1. The Descriptive Statistics*

Table 1 presents the descriptive statistics of the study.

**Table 1.** Descriptive statistics of research variables.

| Sign | Scale | Mean | Minimum | Maximum | Deviation | Skewness | Elongation |
|------|-------|------|---------|---------|-----------|----------|------------|
| OCF | Ratio | 0.152 | −0.681 | 1.147 | 0.167 | 0.789 | 3.299 |
| GROWTH | Ratio | 0.198 | −0.931 | 7.815 | 0.505 | 6.525 | 84.204 |
| R&D | Ratio | 0.000 | 0.000 | 0.016 | 0.001 | 9.100 | 97.422 |
| INTAN | Ratio | 0.006 | 0.000 | 0.145 | 0.011 | 4.833 | 38.920 |
| PPE | Ratio | 0.254 | 0.000 | 0.857 | 0.182 | 0.945 | 0.436 |
| NOL | Dummy | 0.111 | 0.000 | 1.000 | 0.315 | 2.469 | 4.104 |
| AGE | Logarithm | 3.545 | 1.945 | 4.158 | 0.403 | −0.827 | 0.023 |
| LEV | Ratio | 0.453 | 0.005 | 0.977 | 0.225 | 0.150 | −0.871 |
| MTB | Ratio | 1.722 | −277.241 | 121.509 | 11.619 | −14.144 | 329.579 |
| ROA | Ratio | 0.086 | −0.486 | 0.477 | 0.090 | −0.227 | 2.458 |
| SIZE | Logarithm | 13.603 | 9.614 | 18.936 | 1.448 | 0.543 | 0.940 |
| BUSEG | Ratio | 1.931 | 3.617 | 355.810 | 11.820 | 23.805 | 663.459 |
| FIRMVALUE | Ratio | 0.855 | −0.185 | 7.078 | 0.700 | 1.993 | 7.525 |
| TA | Million Rial | 81,922.272 | −12,967,990 | 33,213,308 | 1,579,902.5 | 8.516 | 187.305 |
| BTD | Ratio | 0.138 | −2.443 | 0.705 | 0.174 | −2.258 | 37.224 |
| Ms | Ratio | 0.038 | 8.867 | 0.564 | 0.069 | 3.788 | 17.018 |
| ASSET | Million Rial | 3,901,895.3 | 24,012 | 180,164,197 | 13,207,974 | 7.052 | 60.381 |
| FCF | Ratio | 0.194 | −1.957 | 1.326 | 0.243 | 0.308 | 6.081 |
| Efficiency | Dummy | 0.263 | 0.000 | 1.000 | 0.440 | 1.072 | −0.850 |
| TA/ASSET | Ratio | 0.040 | −0.744 | 1.201 | 0.169 | 1.466 | 7.564 |

The variables' minimum and maximum values indicate the amount of available data collected from companies. In the descriptive statistic, the skewness coefficient is the index for measuring the parameters of asymmetry deviation. The negative value of skewness in variables of age by −0.827, growth opportunity (MTB) by −14.144, return on assets (ROA) by −0.227, and tax avoidance (BTD) by −2.258 indicates that the asymmetrical distribution has a skewness toward a smaller value (negative skewness). Positive skewness in other variables shows that the asymmetrical distribution skews toward the bigger value (positive skewness). The skewness coefficient is the society dispersion measurement index to the normal distribution. The negative value of the skewness coefficient in the research variables, including financial leverage (LEV) by −0.871 and firm efficiency (EFFICIENCY) by −0.850, shows that research variables of society distribution are shorter than the normal distribution, so their dispersion is more than the normal distribution. The positive value of the skewness coefficient in other study variables reveals that research variables of society distribution are longer than the normal distribution. Hence, their dispersion is less than the normal distribution.

### 4.2. Inferential Statistics

First, it is necessary to fit the model using the integrated data method, ordinary least squares, or panel data, for which the F-Limer test is used. The null hypothesis (H0) expresses no difference between estimated coefficients for every cross-section and cumulatively estimated coefficients. This means it is unnecessary to fit the model using the panel data. In other words, an integrated data model or ordinary least squares has more priority over the fixed effects model. After performing the F test, the calculated F is compared with the critical value of the F statistic. If the probability value of the calculated F statistic is less than 0.05, the null hypothesis is rejected and necessary to fit the model using the panel data model. Table 2 indicates the results of the F test.

According to Table 2, the F-Limer test shows that the value of obtained statistics used for testing the research hypothesis is equal to 4.413 and 5.247, respectively. It is bigger than the critical value of the statistic at the 95% level. Given the obtained probability value of the test that is less than 0.05, the null hypothesis, namely, the priority of ordinary least squares, is rejected, and the panel data model is accepted.

**Table 2.** The results of the F-Limer test performed for selecting ordinary least squares or panel data.

| Model | Null Hypothesis | Test | Statistic | Probability | Result |
|---|---|---|---|---|---|
| 1 | The priority of ordinary least squares | F-Limer | 4.413 | >0.001 | The null hypothesis is rejected |
| 2 | The priority of ordinary least squares | F-Limer | 5.247 | >0.001 | The null hypothesis is rejected |

The Hausman test is based on the presence or absence of a relationship between the error of estimated regression and independent variables. If such a relationship exists, it is the random effects model; otherwise, the fixed effects model is applicable.

This test statistic has a chi-square distribution with a K-1 degree of freedom (K-1 equals the Xs). If the value of the calculated chi-square is more than the critical value of the chi-square in the table or, in other words, if the *p*-value of this test is less than the defined alpha (5%), the H0 is rejected. This means that the fixed-effects model is better than the random-effects model. The results of the Hausman test regarding the model of the first hypothesis are depicted in Table 3.

**Table 3.** The conducted Hausman test for determining the random effects model against the fixed effects model.

| Model | Null Hypothesis | Test | Statistic | Probability | Result |
|---|---|---|---|---|---|
| 1 | Use of random effects method | Hausman | 28.939 | 0.001 | The null hypothesis is rejected |
| 2 | Use of random effects method | Hausman | 1295.900 | 0.000 | The null hypothesis is rejected |

Given the results of Table 3 obtained from the Hausman test and before testing the research models, the achieved statistic value from the test in models is equal to 28.939 and 1295.9, respectively. Given the probability value obtained from the test, which is less than 0.05 in the models, the H0 is rejected, that is, the priority of the fixed-effect method in the year–company under study to test the hypotheses.

*4.3. Evaluating the Classic Hypotheses in the Research Models*

The classic models of linear regression have a set of hypotheses called the classical hypotheses. Among the proposed hypotheses, the normality of errors, lack of serial autocorrelation, and variance homogeneity are of the utmost importance. Any defect in the hypotheses mentioned above could pose different problems for the estimated regression, damaging the results. If even one of these hypotheses is not set, model estimation should be carried out using the generalized method of moments.

The linear regression model is the normality of error distribution. The classic regression cannot be employed if such a hypothesis is not developed. According to Table 4, the Jarque–Bera test is used to test the normality of errors. If the statistic probability is more than 5%, the H0 concerning the normality of error distribution is accepted.

**Table 4.** The conducted Jarque–Bera test for establishing the random effects model against the fixed effect model.

| Model | Null Hypothesis | Test | Statistic | Probability | Result |
|---|---|---|---|---|---|
| 1 | The normality of error distribution | Jarque–Bera | 3.391 | 0.082 | The null hypothesis is confirmed |
| 2 | The normality of error distribution | Jarque–Bera | 3.919 | 0.053 | The null hypothesis is confirmed |

### 4.4. Analyzing the Hypothesis of Serial Autocorrelation among Errors

According to Table 5, the null hypothesis of the Breusch–Godfrey test suggests that there is no serial autocorrelation among errors. Since the *p*-value of the test is more than 5%, there is no variance heterogeneity, and the null hypothesis is accepted.

**Table 5.** The results of the Breusch–Godfrey test to diagnose serial autocorrelation among errors.

| Model | Null Hypothesis | Test | Statistic | Probability | Result |
|---|---|---|---|---|---|
| 1 | Lack of serial autocorrelation | Breusch–Godfrey | 1.124 | 0.324 | The null hypothesis is confirmed |
| 2 | Lack of serial autocorrelation | Breusch–Godfrey | 2.601 | 0.207 | The null hypothesis is confirmed |

Given the Breusch–Godfrey test results, the null hypothesis is not rejected at a 5% error level, so there is no serial autocorrelation among errors in the regression.

### 4.5. Analyzing the Hypothesis of Homogeneity of Variance of Errors

According to Table 6, the null hypothesis of the Breusch–Pagan test suggests no variance heterogeneity. Since the *p*-value of the test is more than 5%, there is no variance heterogeneity, and the null hypothesis is accepted.

**Table 6.** The results of the Breusch–Pagan test to diagnose the homogeneity of variance of errors.

| Model | Null Hypothesis | Test | Statistic | Probability | Result |
|---|---|---|---|---|---|
| 1 | Variance homogeneity between errors | Breusch–Pagan | 3.375 | 0.058 | The null hypothesis is confirmed |
| 2 | Variance homogeneity between errors | Breusch–Pagan | 3.375 | 0.115 | The null hypothesis is confirmed |

Given the results obtained from the Breusch–Pagan test at a 5% error level, the null hypothesis is not rejected, so there is no variance heterogeneity in the regressions.

### 4.6. Evaluating Research Models

In this section, we fit the models using the regression test after establishing the diagnostic tests in observations and evaluating the classic hypotheses in the upcoming models. We describe the research model to test the first hypothesis in Table 7.

According to the first hypothesis results in Table 8, we expect a significant negative relationship between managerial ability and tax avoidance. The results of Table 7 show the regression model estimation, suggesting at a 5% of error level with a coefficient of ($-1.223$) and value of (*p*-0.000), there is a negative relationship between management ability and tax avoidance. In other words, increasing the management ability index would lead to lower tax avoidance. Given this inverse relationship between managerial ability and tax avoidance, the null hypothesis is confirmed at 95%, and the first hypothesis is accepted.

**Table 7.** The results of the estimation of the first regression model coefficients using the fixed panel method.

| Variable | Coefficient (β) | Standard Deviation | t Statistic | *p*-Value |
|---|---|---|---|---|
| C | 1.317 | 1.024 | 1.286 | 0.000 |
| MA | −1.223 | 2.700 | −0.453 | 0.000 |
| ROA | 1.300 | 1.244 | 1.045 | 0.000 |
| LEV | −1.801 | 5.488 | −0.328 | 0.001 |
| NOL | −1.145 | 2.897 | −0.395 | 0.000 |
| PPE | −1.950 | 6.729 | −0.289 | 0.003 |
| INTAN | −7.842 | 8.891 | −0.882 | 0.377 |
| SIZE | −4.348 | 1.746 | −2.490 | 0.129 |
| MTB | 1.583 | 5.613 | 0.282 | 0.004 |
| AGE | 8.775 | 1.164 | 7.538 | 0.451 |
| R&D | −2.160 | 1.049 | −2.059 | 0.836 |
| Industry2 | 3.782 | 3.735 | 1.013 | 0.199 |
| Industry3 | 1.758 | 3.379 | 0.520 | 0.000 |
| Industry4 | −9.716 | 3.333 | −2.915 | 0.003 |
| Industry5 | 4.406 | 3.478 | 1.267 | 0.205 |
| Industry6 | −2.473 | 3.437 | −0.719 | 1.060 |
| Industry7 | −1.852 | 3.601 | −0.514 | 0.607 |
| Coefficient of determination | 0.233 | F statistic | | 33.119 |
| The adjusted coefficient of determination | 0.198 | *p*-value | | 0.000 |

Note: all the variables are explained in the previous section.

**Table 8.** The results of the estimation of second regression model coefficients using the fixed panel method.

| Variable | Coefficient | Standard Deviation | t Statistic | *p*-Value |
|---|---|---|---|---|
| C | 0.564 | 0.204 | 2.764 | 0.005 |
| TAXAVOID | −0.067 | 0.016 | −4.188 | 0.000 |
| MA | −0.006 | 0.051 | −0.118 | 0.897 |
| TAXAVOID * Mahi | −0.048 | 0.021 | −2.286 | 0.022 |
| ROA | −4.593 | 0.249 | 18.446 | 0.000 |
| LEV | −4.153 | 0.105 | −39.552 | 0.000 |
| NOL | 0.010 | 0.055 | 0.182 | 0.842 |
| PPE | 0.215 | 0.128 | 1.679 | 0.093 |
| INTAN | 0.312 | 1.721 | 0.181 | 0.035 |
| SIZE | 0.072 | 0.034 | 2.117 | 0.126 |
| MTB | 0.0015 | 0.001 | 1.50 | 0.000 |
| AGE | −0.778 | 0.227 | −3.427 | 0.000 |
| R&D | 25.331 | 19.952 | 1.269 | 0.204 |
| OCF | 0.084 | 0.115 | 0.730 | 0.462 |
| GROWTH | 0.149 | 0.025 | 5.96 | 0.000 |
| Industry2 | −0.640 | 0.074 | −0.865 | 0.000 |
| Industry3 | −0.760 | 0.067 | −11.343 | 0.000 |
| Industry4 | −0.171 | 0.066 | −2.591 | 0.009 |
| Industry5 | −0.215 | 0.069 | −3.116 | 0.001 |
| Industry6 | −0.177 | 0.068 | −2.603 | 0.009 |
| Industry7 | −0.449 | 0.071 | −6.324 | 0.000 |
| Coefficient of determination | 0.669 | F statistic | | 156.664 |
| The adjusted coefficient of determination | 0.566 | *p*-value | | 0.000 |

According to the second hypothesis, we expect a negative relationship between tax avoidance and firm value. The second regression model and variable coefficient of TAX-AVOID are used for testing the second hypothesis. After confirming the fixed effects model,

the tests estimated the regression model (2). As mentioned earlier, it is vital to test the significance of the entire model before evaluating variables and the acceptance/rejection of the hypotheses, which is possible through the calculation of the F statistic and *p*-value. Since the calculated probability value for this statistic is less than 0.05, we could confirm the significance of the entire model with 95% of confidence. The significance of the entire model was confirmed using the probability value of the Fisher statistic (0.000). The regression results of the variable TAXAVOID, at a 5% error level with a coefficient of ($-0.067$) and *p*-value of ($p$-0.000), indicate a significant and negative relationship between tax avoidance and firm value. In other words, the increase in tax avoidance would lead to a decrease in firm value. Thus, the null hypothesis is confirmed with 95% confidence, and the second hypothesis is accepted.

Moreover, according to the specifications of the third hypothesis, we expect a moderating role of managerial ability in the association between tax avoidance and firm value. After confirming the fixed effects model, the regression model (2) was estimated given the conducted tests. Considering the F statistic and *p*-value of the whole model, we can confirm the significance of the entire model with 95% confidence using the probability value of the Fisher statistic (0.000). The regression results of the TAXAVOID*Mahi variable, at a 5% error level and coefficient of ($-0.048$) and *p*-value of ($p$-0.022), indicate that higher ability managers play an ameliorating role in decreasing the negative impact of tax avoidance on the firm value. In other words, in companies with highly ability managers, the intensity of the inverse relationship between tax avoidance and firm value declined. Thus, the null hypothesis is confirmed with 95% confidence, and the third hypothesis is accepted.

## 5. Discussion and Conclusions

The main objective of this study is to assess the potential impact of managerial ability on employing tax avoidance strategies. We further examine the overall influence of such strategies on firm value and the role of the managers' ability to rectify the negative impact of conducted strategies on firm value.

The findings related to the first hypothesis state that there is a significant and negative relationship between managerial ability and tax avoidance. In other words, higher-ability managers tend to invest in other projects to increase the firms' value compared to tax avoidance strategies. Table 9 illustrates the results of the hypotheses.

**Table 9.** The comparison between hypotheses and results.

| No. | Hypotheses | Results |
|---|---|---|
| H 1 | Managerial ability has a negative impact on employing tax avoidance strategies | Negative and significant |
| H 2 | Employing greater tax avoidance strategies decreases the firm value. | Negative and significant |
| H 3 | Managers can mitigate the negative association between tax avoidance and firm value. | Negative and significant |

The obtained results are in line with the previous studies, similar to Hanlon and Heitzman (2010) and Rego and Wilson (2012), Park et al. (2016), and Akbari et al. (2018). Moreover, the findings articulate a negative relationship between tax avoidance and firm value. In this regard, Park et al. (2016), Sikes and Verrecchia (2016), Jacob and Schütt (2019) and Hutchens et al. (2019) propose similar findings.

Finally, we find that managerial ability plays a mitigating role in the negative association between tax avoidance and firm value. It means that high-ability managers can employ an optimum level of tax avoidance strategies. The papers such as Park et al. (2016), Baik et al. (2018), Gul et al. (2018), Khurana et al. (2018), Ratu and Siregar (2019), Lee and Yoon (2020), Salehi et al. (2020b), also find an ameliorating role for managerial ability in mitigating negative factors on firm value

The findings suggest that tax avoidance procedures maximize the owners' wealth and decrease the firms' value under their management. Therefore, considering our results, they

may evaluate the outcome of their decisions, more precisely, by concentrating on available strategies such as tax avoidance and other projects. For equity owners and investors, the findings argue that employing high-ability managers might be one of the most important preferences as a governing element because such a factor plays an undeniable role in employing an optimum level of different strategies, including tax avoidance.

This paper explains the effect of managerial ability on mitigating potential negative factors on firm value. Future studies might explore the influential factors on managerial ability. For instance, which factors matter, managers' personalities such as narcissism, conservatism, hubris, or managers' qualifications, including educational level, background work experience, industry specialization?

**Funding:** The paper recives no external faunding.

**Institutional Review Board Statement:** Not applicable.

**Informed Consent Statement:** Not applicable.

**Data Availability Statement:** The data will be available at request.

**Conflicts of Interest:** The author has no conflict of interest.

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
