# Peer review of "The Effectiveness of Management Ability on Firm Value and Tax Avoidance"

_jrfm, doi:10.3390/jrfm15110539_

Round 1

Reviewer 1 Report

I recommend it for publication

Author Response

Dear Reviewer,

Thank you very much for your effort

Reviewer 2 Report

Dear authors,

In the abstract add the theme of the article because you begin with the goal. Why is this theme important for readers?

In the part of the Discussion please add a table with hypotheses and their results.

Statistic analysis is very excellent.

Thank you, this article was very interesting.

Author Response

Dear Reviewer,

Thank you very much for your attention and for evaluating the paper, based on your good comments, the following issues are added to the paper:

In the abstract add the theme of the article because you begin with the goal. Why is this theme important for readers?
This part is revised; thank you
In the part of the Discussion please add a table with hypotheses and their results.
This part is added to the paper

Reviewer 3 Report

The paper presents an interesting topic, the one of investigation of the relationship between tax avoidance, management ability, and firm value. Three hypotheses are proposed to meet the objective of the paper, the data being gathered from the official website of the Tehran Stock Exchange during 2014-2020.

The paper is well-written, however, there are some misspelling or style errors that should be corrected in order to improve the overall quality of the paper presentation.

-        The sentence does not end (Line 572);

-        Ending points missing (Line 572, 579);

-        The equations should be numbered;

-        A note under Table 1, Table 7, Table 8 regarding the explanation of the variables used should be included, in order to make the reading of tables more appropriate, without recurring to previous tables and text;

-        A proof-reading of the paper by a native speaker is recommended (a reformulation of some sentences like: “we can conclude an inverse relationship between managerial ability and tax avoidance”

-    

Author Response

Dear Reviewer,

Thank you very much for your good comments on the paper, further all comments are incorporated in the paper as follows:

The paper is well-written, however, there are some misspelling or style errors that should be corrected in order to improve the overall quality of the paper presentation. Thank you! The proofreading is conducted on the paper

-        The sentence does not end (Line 572); Corrected

-        Ending points missing (Line 572, 579); Corrected

-        The equations should be numbered; Added

-        A note under Table 1, Table 7, Table 8 regarding the explanation of the variables used should be included, in order to make the reading of tables more appropriate, without recurring to previous tables and text; added

-        A proofreading of the paper by a native speaker is recommended (a reformulation of some sentences like: “we can conclude an inverse relationship between managerial ability and tax avoidance”  corrected